# Biological Degradation of Plastics and Microplastics: A Recent Perspective on Associated Mechanisms and Influencing Factors

**DOI:** 10.3390/microorganisms11071661

**Published:** 2023-06-26

**Authors:** Zeming Cai, Minqian Li, Ziying Zhu, Xiaocui Wang, Yuanyin Huang, Tianmu Li, Han Gong, Muting Yan

**Affiliations:** 1College of Marine Sciences, South China Agricultural University, Guangzhou 510641, China; 17820577242@163.com (Z.C.); koitoyuu045@foxmail.com (M.L.); zz2545440446@163.com (Z.Z.); cc45683968exo@163.com (X.W.); 17666057810@163.com (Y.H.); 13503034564@163.com (T.L.); 2Department of Civil and Environmental Engineering, The Hong Kong Polytechnic University, Hung Hom, Kowloon, Hong Kong

**Keywords:** plastics, microplastics, biodegradation, microorganism, enzymes, pretreatment

## Abstract

Plastic and microplastic pollution has caused a great deal of ecological problems because of its persistence and potential adverse effects on human health. The degradation of plastics through biological processes is of great significance for ecological health, therefore, the feasibility of plastic degradation by microorganisms has attracted a lot of attention. This study comprises a preliminary discussion on the biodegradation mechanism and the advantages and roles of different bacterial enzymes, such as PET hydrolase and PCL-cutinase, in the degradation of different polymers, such as PET and PCL, respectively. With a particular focus on their modes of action and potential enzymatic mechanisms, this review sums up studies on the biological degradation of plastics and microplastics related to mechanisms and influencing factors, along with their enzymes in enhancing the degradation of synthetic plastics in the process. In addition, biodegradation of plastic is also affected by plastic additives and plasticizers. Plasticizers and additives in the composition of plastics can cause harmful impacts. To further improve the degradation efficiency of polymers, various pretreatments to improve the efficiency of biodegradation, which can cause a significant reduction in toxic plastic pollution, were also preliminarily discussed here. The existing research and data show a large number of microorganisms involved in plastic biodegradation, though their specific mechanisms have not been thoroughly explored yet. Therefore, there is a significant potential for employing various bacterial strains for efficient degradation of plastics to improve human health and safety.

## 1. Introduction

Plastics have caused a great deal of ecological problems that have attracted a lot of attention worldwide. Plastic products are useful and they bring convenience to our lives. However, as a persistent pollutant, they can remain in natural environment for hundreds of years or longer [1]. In 1950, the global production of plastics (excluding fiber) was 1.3 million tons, and by 2018, it reached an alarming level of 359 million tons (excluding fiber), resulting in widespread environmental contamination [2]. It is estimated that only 10% of plastic waste is recycled, 14% is incinerated, and the rest is dumped into landfills, ultimately entering the natural environment [3]. Plastics are transported through air and water currents, therefore, they not only affect the natural environment but also pose adverse ecological impacts in the deep-sea sediments and polar regions [4]. In fact, plastics and microplastics (MPs, size 5 mm) have been commonly found in freshwater, the atmosphere, and soil environments across the world. A class of high molecular weight polymers is referred to as plastics in general. The formation of “white pollution” has caused serious environmental pollution and ecological damage around the world. Polyethylene (PE), polypropylene (PP), polyethylene terephthalate (PET), polystyrene (PS), and polyvinyl chloride (PVC) are the most widely utilized plastics, with PE and PP making up more than half of global output [5]. Plastic waste is persistent when released into the environment, high polymers may take a much longer time for plastics to degrade in the environment, and even complete degradation of plastics may take centuries [6]. Therefore, it is necessary to investigate the pathways of plastic degradation and their efficiency in the environment.

Plastics are made up of numerous compounds, including a vast range of chemicals. These include basic substances, such as monomers, oligomers, polymers, and additives, and the additives are mainly divided into plasticizers, antioxidants, heat stabilizers, and pigments [7]. Apart from the pristine additives, plastics adsorb chemical pollutants from the surrounding environment, which makes plastics a hub for toxic substances. The microbial degradation of such plastics containing additives and adsorbed chemicals also needs special attention [8]. MPs contain additives with a large amount of mixed chemical components that may bio-accumulate in the food chain [9]. When released into the environment, plastics and MPs can undergo degradation processes, such as hydrolysis, thermal oxidative degradation, biodegradation, electrochemistry, and photodegradation [10]. Microbial biodegradation has been regarded as one of the main ways to effectively deal with plastic pollution [11]. It is a process in which microorganisms use carbon sources in the form of organic matter to metabolize, which can not only produce non-toxic by-products but also provide energy to microorganisms or transform them into other useful products [12]. Biodegradation may effectively reduce the harm of additives to plastics and achieve a better ecological environment [13].

Plastics likely have a negative impact on biodiversity loss and the life cycle of plastics and plastic pollution may directly change. Microbial degradation is of great significance in reducing the negative impact of plastics. Due to the diversity and richness in the composition of the microbial communities, there will be more possibilities for plastic biodegradation research. We focus on the advantages of biodegradation, especially in terms of bacterial degradation. The literature suggests that the more suitable and eco-friendly way lies in microbial-mediated degradation [9]. Microbial enzymatic degradation will be based on the degradation ability of bacterial enzymes and appropriate biodegradation efficiency [14]. Although knowledge of the environmental degradation of plastics is still limited, the fate and effects of plastics are determined by the type and strength of degradation [2]. Therefore, exploring the ability of bacteria and the interaction between bacterial enzymes and plastics is critical to obtain key biodegradable microorganisms. There are not many in-depth studies on biodegradation, which can be looked at in different ways depending on the additives in plastics and the enzymes in bacteria. This article’s goal is to examine the viability of biodegradable plastics, with a particular emphasis on the most current research on the functions of various microorganisms and their enzymes in the biodegradation of synthetic plastics. It also pointed out the existing microbial degradation challenges and corresponding methods. Emphasis has also been placed on the biodegradation mechanism and the degradation ability and effectiveness of microbial enzymes, and finally, the importance of pretreatment for biodegradation.

## 2. The Challenges and Feasibility of MPs/Plastics in Biodegradation

### 2.1. The Challenges in Biodegradation of MPs/Plastics

MPs/plastics are not part of rich functional groups and hydrolyzable bonds, they are not a suitable substrate for microbial attachment and enzymatic reaction [15] due to their intrinsic composition and properties, such as durability and resistance to degradation. The recalcitrant polymers do not completely degrade, making these polymers difficult to assimilate in the environment [16]. The recalcitrant polymers with less flexibility and long chains effectively biodegrade into small monomers, which require more enzymatic activities for a longer period of time to cause specific sensitive bonds with the chemical groups on the polymer side chain or polymer chain and enhance their fracture [17]. In a biodegradation process, the problem of conventional petroleum-based plastics involves not being completely decomposed and assimilated by microorganisms when they fragmented under abiotic factors over a long time, such as UV radiation, temperature, and physical stress [4]. Because plastic is a long-chain molecule or macromolecule with a single functional group, high molecular weight (MW), hydrophobicity, and crystallinity, the ability of microbes to break it down has been significantly reduced as a result of its manufacturing performance. For example, plastics without pro-oxidants lack functional groups/enzyme-sensitive ester bonds that may not often be subjected to enzymatic attacks. On the contrary, biodegradable plastics might have appropriate functional groups or enzyme-sensitive ester linkages that could be attacked by enzymatic, photo-, and thermo-oxidation [18]. High polymers are more difficult to disintegrate in comparison to other plastics, such as emerging biodegradable plastics, due to their short chains, low molecular weights, and flexible, abundant functional groups [18]. The problem and difficulty of plastic degradation depend on traditional plastics with a unique chemical composition. Therefore, microbial degradation ability and efficiency should be improved, or more suitable and efficient strains should be found [3]. Currently, there are still undeveloped technologies and cost issues, such as technological development and capital consumption not in line with social development. Microorganisms are difficult to efficiently degrade plastics, and there are more possibilities for microorganisms to search for and match plastic degradation [19].

Moreover, one of the reasons why plastic degradation is currently a hot environmental research topic is that turning plastics into small particles can release toxicity in this difficult task. First, phthalates (PAEs) are widely used as plasticizers to introduce unique characteristics in plastic products, such as plasticity, flexibility, and durability. Humans are exposed to PAEs mainly through food, and they can cause serious problems, such as endocrine disruption, metabolic disorders, and reproductive toxicity [20]. There is also a known endocrine disruptor compound that accounts for a large proportion of synthetic polymers and additive plasticizers, Bisphenol A (BPA). It is produced in large quantities worldwide, mainly used in the production of various polymer materials and also in polycarbonate synthesis [21]. Plastics are macromolecular substances which initially show relatively little toxicity and will not be absorbed by the human body. The peak of plastic toxicity is due to the toxic additives that need to be added during processing or the insufficient reaction during synthesis. However, small particulate compounds, such as monomers, are highly toxic [22] (Figure 1). The properties of MPs and POPs, and environmental conditions are the factors responsible for the adsorption of POPs on MPs [23]. The toxicity or harm may rise in the monomer stage when a large amount of POPs is adsorbed on MP surface, the efficiency of transfer of the ingested POPs in the food chain is unknown, but the range is wide [24]. Human exposure to monomers and POPs can cause disrupting the natural gut microbiota balance through inhalation, ingestion, and skin contact (Figure 1). The toxicity and pollution range of plastics and MPs are wide, and they affect ecology and humans in all aspects. We urgently need to find the reasons for the biodegradation by microorganisms.

### 2.2. The Feasibility in Biodegradation of Plastics/MPs

Microbial diversity is the key condition for biodegradation. Microorganisms include bacteria, fungi, and a few algae, which have powerful functions and abilities, such as small size, fast absorption, and metabolism, strong adaptability, easy variation, and wide distribution [5,25]. Microorganisms have congenital or acquired abilities to survive even in places with particularly harsh environments. The microbial communities can adapt to new environments and attach to them, and they mainly release certain enzymes that allow them to utilize persistent plastic pollution as their only carbon source [26]. Biodegradation is influenced by a combination of biological and abiotic factors. Biological factors include bacteria, fungi, biofilms, and microbial communities as the carbon source food for microorganisms. Abiotic factors are from several aspects of natural photooxidation and man-made physical and chemical degradation. These factors catalyze complex metabolic reactions to form the ecosystem structure and function of biodegradable plastics [27]. The microbial degradation of plastics and MPs is affected by the growth of microorganisms and factors related to the external environment. Environmental variables and the habitat under study may be major determinants of microbial diversity and richness [28]. In addition to the microorganisms involved, the circumstances for plastic biodegradation also depend on the surface properties and surface structure of plastic materials, such as roughness, electrostatic interaction, topography, hydrophobicity, and free energy [29].

More possibilities for effective biodegradation of polymers involve various microorganisms. In marine habitats, biofilms formed by microbial populations are conducive to plastics and MPs degradation. The formation of microbial biofilm is the way of polymer biodegradation. It is initially attached by microorganisms and forms microbial biofilm on the surface of the plastic, which is called a plastisphere [30]. Microbial communities are associated with biofilms. The biofilms contain different microbial communities and contain potential plastics degrading microorganisms. For instance, the hydrophobicity of PE interferes with the formation of biofilm. However, the micro-bacterial community with high cell surface hydrophobicity can better adhere to the surface of PE and form biofilm on its surface [31]. In nature, however, various studies have shown that the bacteria typically collaborate in communities. The microbial community has a variety of microorganisms to participate in the degradation, which improves efficiency compared to a single microorganism. Microbial communities or consortia can degrade complex compounds into single monomers due to the observation of multiple microorganisms that coexist in an environment to function together [5].

The pure cultures of bacteria isolated from most environments show that bacteria have diversity and functionality. Through the identification of a large amount of data, species belonging to various bacterial phyla, including *Proteobacteria*, *Firmicutes* and *Actinobacteria*, are able to degrade plastics. Most potential bacteria that can biodegrade plastics were isolated from contaminated sites, such as landfills [32]. Based on bacteria’s inherent ability to break down long-chain fatty acids, plastic degradation is possible. Bacterial species from *Pseudomonas*, *Escherichia*, and *Bacillus* genera exhibit enormous potential for degrading plastics, especially for recalcitrant polymers, such as PE, PET, PS. Additionally, it has been demonstrated that bacterial strains, such as *Pseudomonas aeruginosa*, *Bacillus megaterium*, *Rhodococcus ruber*, and others may break down the thermoplastics PE and PET [5]. Furthermore, fungi able to degrade plastics have a powerful enzymatic system. Filamentous fungi play an important role in the degradation and mineralization of plastic pollutants. They have the ability to degrade PE and PET [33]. With more research, it was discovered that microorganisms able to degrade plastic vary in the diversity, abundance, and activity, providing plastic-degrading evidence.

## 3. Mechanism of Plastics/MPs Biodegradation

Microorganisms depolymerize MPs, which leads to extracellular enzymes and free radicals produced by the microbes to catalytically cleave biodegraded plastics into smaller elements. Microbiological degradation of plastics is generally a process of breaking down the polymer into shorter chains or smaller molecules (e.g., oligomers, dimers, and monomers). The polymers that are tiny enough are depolymerized into monomers that may be necessary for microbial uptake and growth to pass through semipermeable membranes and eventually mineralization in cells. Then, the monomers in the cells are either mineralized into CO_2_, H_2_O (under aerobic conditions) or CO_2_, H_2_O and CH_4_ (under anaerobic conditions) to produce biomass for energy (Figure 2). 

Microbial colonization on plastic surface through adhesion or exposure is the initial step for microbial degradation of plastics, which is the premise of enzymatic degradation. The second step is hydrolysis, which comprises the combination of enzyme and polymer matrix, and then catalytic hydrolysis and cracking. The bound enzyme is a hydrolase, which catalyzes the hydrolysis of organic matter [34]. The extracellular and intracellular enzymes from microorganisms involve two important processes. Extracellular enzymes, such as depolymerases and hydrolases, induce hydrolytic cleavage to the polymer chain. The polymer chain after being attacked by enzyme enhancement results in small oligomers or monomers that bacteria can integrate with the cell. Additionally, small oligomers or monomers can be absorbed in the cell after their metabolism by enzymes in the cell [25]. Compared with oxidative degradation for large plastics or plastic fragments, enzymatic degradation can be measured by weight loss and additions of functional groups. One of the biggest changes in MPs is the weight loss of the plastics during biodegradation. This is the initial MPs weight and the weight reached after contact with microorganisms [32].

Something produced by microbial degradation and metabolism is conducive to the environment. Polymer morphology, such as crystallinity, is a key factor affecting biodegradability, which greatly affects its biodegradation rate. The reason is that the enzyme mainly attacks the amorphous domain of the polymer, and its molecules are loosely arranged and easier to degrade [34]. Therefore, microbial enzymes play a key role in environmental remediation and ecological health by biodegradation, participate in bio-transformation and biodegradation processes, and combine the advantages of efficiency and functionality, making the bio-catalytic processes more competitive, clean, safe, and eco-friendly [13].

## 4. Microorganisms and Their Enzymes in Plastics/MPs Degradation

### 4.1. Microbial Degradation of Commonly Used Plastics/MPs 

The degradation evidence can be mainly decided by three results of technology: (1) Changes in the plastics/MPs structure, (2) physical loss of plastics/MPs mass, (3) the generation of plastics/MPs metabolites. These combined results are likely the strongest evidence for plastics/MPs biodegradation [3].

#### 4.1.1. Microbial Degradation of the Commonly Used Recalcitrant Plastics/MPs

Nonhydrolyzable plastics’ primary molecular chain is entirely made up of C-C bonds, as well as PE, PP, PVC, and PS. As nonhydrolyzable plastics, their application range is larger, but the level of biodegradation is relatively low. The nonhydrolyzable MPs are often converted into small molecules of organic matter through redox reactions by bacteria or fungi before being utilized by microorganisms [19]. PA and PET are hydrolyzable plastics that are relatively easy to decompose. There is an increasing example showing that microorganisms can have a corresponding ability to degrade plastic pieces. PET is the most common thermoplastic plastic, with the largest production proportion of almost 70 million tons annually worldwide, mainly used as fiber in textiles and packaging, and a small part is used in film and engineering fields [35]. Currently, bacteria and fungi for the partial degradation of PET are only a few. Therefore, it appears that only a few bacterial phyla exhibit PET degradation features, and the majority of bacterial isolates that can degrade PET are found in the Gram-positive phylum *Actinobacteria* [36], which includes the genera *Thermobifida* and *Thermomonospora* [37]. In the database, several signature enzymes have been reported for PET metabolism, such as PETase, MHETase, TPA dioxygenase, and PCA dioxygenase [38]. *Ideonella sakaiensis* 201-F, a new bacterium that can utilize PET as its main energy as well as carbon source and produce morphological changes in the PET film, was discovered by the researchers after they screened microbial communities from samples contaminated with debris. The discovery of this bacterium is unique in that it can use plastics as its only food source. Producing two enzymes from the strain has the ability to hydrolyze PET, converting PET efficiently into environmentally benign monomers. *Isakaiensis* secretes a PET hydrolase (PETase)that can efficiently hydrolyze PET [38]. *Isakaiensis* strain, like some bacteria, secrete enzymes to hydrolyze MPs and mineralize them in cells [34]. Actually, the whole process is divided into two steps, the strain synthesizes two highly specialized enzymes—PETase and MHETase. However, the exact binding mechanism of PETase is unknown. Additionally, both *Pseudomonas nitroreducens S8* and *Pseudomonas monteilii S17* are capable of colonizing the PET surface as a transparent biofilm layer with a steady population density, allowing them to use PET as a carbon source. The combined use of *P. nitroreducens S8* and *P. monteilii S17*, which produce a PET surface that is highly accessible for PET hydrolases, demonstrated a strong synergistic effect on the disruption of PET surfaces [39].

PS is a high molecular weight synthetic hydrophobic polymer, not easy to be attacked by microorganisms and difficult to degrade, and the degree of PS biodegradation is very low [40]. However, the research has identified the ability for PS degradation in the intestinal tract of mealworms. The mealworm gut microbiome found that two operational taxonomic units (OTUs) *(Citrobacter* sp. and *Kosakonia* sp.) had strong ties with plastics, such as PE and PS, indicating that they have degradation potential for chemically dissimilar plastics. Further studies should be conducted to exploit the gut microbiome of mealworms for the biodegradation of PS [41]. Besides non-hydrolyzable PS, PE biodegradation is also very challenging. PE is a thermoplastic hydrocarbon polymer, with corrosion resistance and electrical insulation properties, and is considered one of the most widely produced and commercialized synthetic polymers worldwide [31]. 

PE contains a carbon-carbon backbone and non-hydrolyzable covalent bonds, its properties are highly resistant to degradation. Microbial consortium may be a relatively effective choice for the degradation of PE in terms of function and inherent synergy [9]. According to reports, *Pseudomonas* sp., *Bacillus* sp., *Mycobacterium* sp., and *Nocardia* sp. are PE-degrading bacteria [42]. Research has shown that a bacterial consortium was capable of forming a thick biofilm on the weathered PE surface and cause changes to the rheological and topographical characteristics of the surface. The amesophilic bacterial consortium obtained from landfill sediment was analyzed. The researchers observed that *Bacillus* sp. and *Paenibacillus* sp., which are both abundant in the bacterial consortium, decreased the mean MPs particle diameter and dry weight after 60 days by a combined 22.8% and 14.7%, respectively [5].

Among PE polymers, high-density polyethylene (HDPE) and low-density polyethylene (LDPE) are the most commonly used polymers, especially LDPE, which has become one of the most important plastic grades for modern polymers. The difference between LDPE and HDPE is that HDPE is a high crystallinity, non-polar thermoplastic resin. The softening point of HDPE is 125–135 °C, while LDPE is 90–100 °C. HDPE is resistant to strong oxidant corrosion, acid, alkali, and various salt corrosion, insoluble in any organic solvent, etc. LDPE is resistant to corrosion in acid, alkali, and salt solutions, but has poor solvent resistance. Due to the production function of HDPE, it is relatively difficult to degrade in terms of microbial degradation [42,43]. In biodegradation, a comparison between bacteria and fungi isolated from the same dump identified that fungi proved more effective for the degradation of PE than bacteria. This study further showed that fungi-treated LDPE films experienced a higher percentage of weight loss than that treated by bacteria [44]. Using fungal and bacterial strains as an individual or as a consortium can efficiently degrade LDPE. The interaction formed ridges and grooves on LDPE, which confirmed the occurrence of degradation [45]. For bacterial development, microorganisms isolated from marine habitats use the carbon in LDPE. After 30 days of incubation, the dry weight loss of LDPE films containing four bacterial isolates was 1, 0, 78, 22, and 0.46%, and after 90 days, it was 1, 1, 72, and 0.97%, respectively. The effectiveness of using LDPE as a carbon source for bacterial growth is determined by calculations of weight loss and carbon mineralization of LDPE film. After 90 days of incubation, the bacterially treated LDPE films’ FTIR spectra revealed new peak generation and shifting of the peak for the functional groups, and SEM images revealed surface deterioration, fragility, damaged layer, cracks, and scratches on all four marine bacteria-treated LDPE films [42]. 

Among the known fungal strains that degrade plastics, *Penicillium oxalicum NS4* (KU559906) and *Penicillium chrysogenum NS10* (KU559907), two fungal strains that have been found to degrade HDPE and LDPE and cause noticeable morphological changes on the plastic sheet, have been isolated from fungi [46]. A prominent strain, *Aspergillus* genus, as the most prominent fungal group, has shown strong plastic-degrading ability. Aspergillus genus degrades synthetic plastics PE and PP, like *A. clavatus*, *A. fumigatus*, and *A. niger* [32]. A plastic dump wasteland isolation of *Aspergillus niger* (ITCC No. 6052) has demonstrated its ability to break down HDPE [47]. Understanding *A. niger* and studying its enzyme system will help to deeply understand its role in HDPE biodegradation. *Aspergillus* sp., including *Rhizopus oryzae* strain *NS5* and *A. niger*, respectively, degrade LDPE and HDPE. In addition, two *Penicillium* strains (*Penicillium oxalicum NS4* and *Penicilliumchrysogenum NS10*) proved to have properties for LDPE and HDPE [48]. The degree of weight loss depended on many aspects, with LDPE and HDPE showing the greatest weight loss in the form of foils, films, or strips. 

#### 4.1.2. Microbial Degradation of Toxic Components in Commonly Used Plastics/MPs 

Since the 1950s, when the mass manufacture of plastics began, the production of all sorts of polymers infused with different additives and fillers has expanded tremendously. PAEs and BPA are prominent plastics that can cause harmful impacts. BPA is toxic to most marine animals and vertebrates. It is potentially carcinogenic because BPA structurally resembles diethylstilbestrol (DES). Therefore, the toxicity is higher when the plastic particles or even smaller stages are exposed [49]. 

The toxic substances, PAEs and BPA, which are some of the main components of plastics, are involved in plastics biodegradation. Most studies on PAE’s biodegradation use bacteria, such as *Pseudomonas*, *Arthrobacter*, *Rhodococcus*, *Bacillus*, *Mycobacterium*, *Delfia,* and *Gordonia* [13]. The microbial degradation of PAEs is through ester bond hydrolysis to produce monoester, which decomposes to produce phthalic acid (PA) and alcohol, and ultimately produces short-chain acid, which would ultimately engage the Krebs cycle [50]. Hydrolase and oxygenase are the enzymes for the biodegradation of PAEs. One of the enzymes, esterase, shows the best activity in degrading phthalates, with pH value and temperature in the range from 7 to 10 and 30 to 70 °C, respectively [51]. It was found that reducing weight is quite effective in the role of pretreatment temperature and hydrolase. An estuary waste electronics dismantling area *Bacillus* sp. GZB isolated facultative anaerobic BPA-degrading bacteria may accelerate BPA biodegradation in water sediment. The addition of *Bacillus* sp. GZB was closely connected to improved biodegradation efficiency. BPA biodegradation was enhanced by adding substance, such as electron donors and co-substrates, or yeast extract, NaCl, humic acid, and glucose, respectively [52]. BPA undergoes bioaugmentation and bio-stimulation in water-sediment microcosms [53]. 

#### 4.1.3. Microbial Degradation of Commonly Used Biodegradable Plastics/MPs

Emerging plastics called biodegradable plastics are prone to degradation, compared to high polymers. Biodegradable plastics (BPs) such as poly-caprolactone (PCL), poly butylene succinate-co-adipate (PBSA), poly butylene succinate (PBS), and poly lactic acid (PLA), when compared to high polymers, are prone to degradation and overcome the environmental pollution and challenges associated with synthetic plastics. Some biodegradable polymers that play a role in composting, such as PCL, polyhydroxybutyrate (PHB), PLA, and PBS, can be degraded significantly. Their chemical structure, which includes ester linkages that make them susceptible to breakdown by microbial enzyme systems, is crucial [15].

Thermophilic actinomycetes with activity to PHB, PCL, or polyethersulfone (PES) were isolated from various environments (like compost) in Taiwan [54]. When exposed to PCL wastes at 50 °C, two thermophilic bacteria isolated from compost demonstrated effective synergy, accelerating PCL breakdown and substantially increasing the amount of the disintegrated polymer [55]. Because PCL is a thermoplastic crystalline polyester, it can be hydrolyzed into small molecules by lipase and then further assimilated by microorganisms. For instance, PCL was completely degraded by *Aspergillus* sp. strain ST-01, isolated from soil after six days incubation at 50 °C by thermotolerant PCL-degrading microorganisms. Catalase and Protease are secreted by *Aspergillus* sp. strain ST-01 as its enzyme. Besides, marine organisms, such as *Pseudomonas*, *Alcanivorax,* and *Tenacibaculum*, isolated from the deep sea sediments, can also effectively degrade PCL [56]. 

### 4.2. Enzymes Involved in Plastics/MPs Biodegradation 

#### 4.2.1. Enzymes Involved in Plastics/MPs Hydrolysis 

A variety of enzymes, including cutinases, esterases, lipases, laccases, peroxidases, proteases, and ureases from bacterial and fungal sources, have been demonstrated to have the capacity to break down PE, PET, and PP [5]. Fungal cellulase systems observed cellulose depolymerization free enzyme to act directly on solid polymeric substrates, and the final step converted monomeric constituents (e.g., cellobiose hydrolysis to glucose) [57]. The properties of fungal enzymes, especially the depolymerases, allow them to break down different polymers, they can generally degrade PET and PE. The distribution and penetrative ability of fungal hyphae are a significant factor in their original colonization before depolymerization and their enzyme capacities to enhance hyphal attachment to hydrophobic substrates. Laccases from *actinomycetes*, *Rhodococcus ruber,* and fungi, such as *Aspergillus flavus* and *Pleurotus ostreatus,* have also exhibited significant degradation of PE. It may be caused by the oxidation of the PE hydrocarbon backbone [5]. Besides, in the biodegradation of conventional low-density plastic LDPE, other fungal species with substantial plastic degrading properties include *Fusarium solani*, *Alternaria solani*, *Aspergillus fumigatus, Spicaria* spp., *Geomyces pannorum*, *Phoma* sp., *Penicillum* spp., etc. [58]. The degradation microorganisms of polyethylene materials mainly include bacteria and fungi, among which there are are mainly fungi such as *F. oxysporum* and *Aspergillus fumigates*. The cultivation of non-degradable plastic waste PE by *F. oxysporum* strains has better tolerance to high concentration PE, which can cause strong oxidation phenomena and observe changes in PE film morphology. *Aspergillus fumigates* form biofilms to degrade LDPE, achieving current experimental efficiency in the fungal degradation of LDPE [59,60].

Some hydrolyzable plastics and PET are degraded by highly efficient degrading bacteria (*Ideonella sakaiensis* 201-F) or hydrolyzed by bacterial enzymes (hydrolase and keratinase), and their enzymes are combined for degradation, such as PETase and MHETase. Previous studies generally used esterases, lipases, keratinases, and other hydrolytic enzymes to degrade PET. When an enzyme specifically hydrolyzing PET is discovered, it is named PETase [61]. A recent study reported the enzymes secreted by bacteria degrading PET polymer step by step, synthesize the two enzymes and act synergistically, increasing the degrading efficiency by six times (Figure 3). PET was hydrolyzed by enzymes of *I. sakaiensis* strain in PET depolymerization into its monomers ethylene glycol (EG) and terephthalic acid (TPA) [62]. During PET depolymerization, TPA and EG in *I. sakaiensis* strain are liberated, which requires PETase and MHETase to act synergistically [57]. Both enzymes’ potential as PET-degrading agents has gained significant attention to reduce plastic waste. PETase effectively impacts the stability of the PET molecules because of its esterase that catalyzes the ester bond. The dimer BHET and monomer MHET are released when PETase hydrolyzes polymeric PET molecules (Figure 3). The MHETase then converts the MHET into TPA and EG [35].

Synergistic enzymes secreted by multiple microorganisms and microbes with a two-enzyme system are the hotspot of the study on the biodegradation of pollutants internally and externally due to their powerful degrading ability and special metabolic type. Similarly, a potential research area is the development of multienzyme systems for the depolymerization of plastic waste. For example, *I. sakaiensis* strain has a dual enzyme system containing PETase and MHETase that has evolved the ability to utilize crystalline polyester substrates. The latest research shows that scientists have designed *I. sakaiensis* strain enzymes to improve degradation efficiency [57]. The research has shown that improved enzymes can allow bacteria to degrade 90% of plastic products within 10 h. Scientists optimize enzymes to transcend natural evolution by genetic engineering, in order to make the enzyme work more efficiently than the natural state, the enzyme needs to be artificially modified. A PET hydrolase that has been refined and tuned has shown great efficiency, producing at least 90% PET depolymerization into monomers during a 10-h period. This enzyme from the bacterium *I sakaiensis* strain 201-F6 outperformed all PET hydrolases reported so far [63]. PET films were virtually completely destroyed by the strain’s two highly specific enzymes, PETase and MHETase, in a couple of weeks. The effective degradation was observed around 30 °C, under mesothermic conditions [35]. PETase and MHETase are two unique enzymes, their concerted action for the hydrolysis of PET is essential. PET degradation relates to the presence of MHETase. Even PETase at low concentrations also improved the degradation efficiency [57]. The enzyme combination with a substrate is closely related to maintaining the structural stability of enzyme biocatalyst [30]. In general, the practices for standard commercial use, such as enzyme immobilization, enzyme entrapping, or encapsulation, hinder the optimal enzyme-substrate interaction in PETase, thereby reducing the overall effect on PET degradation [35]. There is a relatively low turnover rate for PET hydrolases. However, it is enzymatically hydrolyzed at about 65–75 °C, thereby realizing the effective enzymatic degradation of PET through heat-resistant PET hydrolase [36]. The research on PET hydrolase can be expanded to substrate scope of PETase, and PETase variants can be screened to improve the thermal stability and PET degradation ability [27]. Therefore, it is significant to further investigate the molecular mechanism of PET hydrolase and the development and utilization of new PET-degrading enzymes. The studies on PET biodegradation have found multiple cutinase enzymes, which can perform PET depolymerization with the prevalence of esterase enzymes in nature [57]. 

Fungi are members of 10 genera of the *Ascomycota* and *Mucor*, and they degrade polyethylene more quickly than bacteria. The crucial factor causing this distinction is that fungi can adhere to polymer surfaces with hydrophobic surfaces. When using PP and PE as the sole carbon source, it may be possible to culture *Aspergillus*, *Fusarium oxysporum*, and *Penicillium* for three months to test their capacity for degradation. The analysis also validates the presence of biofilms and reveals alterations to PP and PE’s surface. The incubated fungus are also examined, and it is shown that they can survive for more than three months without any extra carbon sources [64]. In *ascomycetes*, *Fusarium oxysporum* can secrete keratinase that can depolymerize PET. Similarly, the cutinase LC-cutinase, which can degrade PET and PCL, was generated from a leaf’s fosmid library and produced in *Escherichia coli*. It may be utilized to modify PET’s surface and break it down. There is a great deal of potential for cutinases to hydrolyze PET and PCL. The underlying mechanisms of cutinase enzyme for degrading specific plastics should be analyzed. In the future, the transformation and cloning of bacteria and enzymes can greatly improve the degradation of various plastics. Recent studies indicated that the two cutinases, LC-cutinase and *Thermobifida fusca* cutinase, achieved higher activity in the experiment, and the residual activity of the latter was higher after 40 h. Thus, LC-cutinase is marginally less stable than *T. fusca* cutinase [65]. In the presence of surfactants and organic solvents, the *T. fusca* cutinases demonstrated a number of benefits, including adaptable hydrolytic activity, good tolerance to surfactants, superior stability in organic solvents, and thermostability. Two types of cutinases with 93% similarity, called TfU 0882 and TfU 0883, were isolated from *T. fusca*, which can metabolize synthetic polyesters. Despite the two enzymes’ remarkable resemblance, only TfU 0883 can break down PET at 60 °C. Although both LC keratinase and BC lip can break down PCL, LC-cutinase has been shown to have a specific PCL-degrading activity of 300 mg/h/mg of enzyme at pH 8.0 and 50 °C, but BC lip’s PCL-degrading activity (8 mg/h/mg of enzyme) was significantly lower than that of LC-cutinase, under the same conditions. When PET was degraded using LC-cutinase, the findings revealed that the degradation rate was 230–970 times more than that of the data for other cutinases [65]. The research report demonstrates that LC-cutinase’s hydrolytic activity is what causes PET to deteriorate. It exerts and enhances specific abilities and can totally break down PET into TPA and EG [66]. With a thorough understanding of these enzymes’ protein structures, it is possible to improve their catalytic efficacy in degrading different types of polymers, including PET. Microorganisms evolved in the process and found suitable enzymes for mutation, selecting mutated keratinase to decompose PET in a short period of time [35]. In order to develop more efficient enzymes, relational studies can analyze the detailed structure and function of cutinase.

#### 4.2.2. Biodegradation Associated Enzymes Produced by Microorganisms in the Extreme Environments

Extreme environmental microorganisms, such as halophiles and psychrophiles, exhibit plastic degradation potential. *Thermophilic*, *alkaliphilic*, *halophilic*, and *psychrophilic* bacteria in multiple extreme environments are potentially capable of degrading synthetic plastics (Table 1). This special type of environment requires screening of more effective degradation bacteria [67]. Actually, plastic contaminated places are characterized by extreme environmental conditions, such as low or elevated temperatures, acidic or alkaline pH, high salt concentrations, or high pressure. With regard to the features of thermophilic and halophilic enzymes, they have a longer life cycle, allowing their storage at room temperature and preventing a significant loss of enzymatic activity [68]. Consequently, as a source of plastic-degrading enzymes and microorganisms, there is a significant scope for thermophile and extremophile microbiomes to be explored. The most thermally stable leaf-branching compost cutinase (LCC) had the highest PET depolymerization rates at 65 °C [3]. At elevated temperatures, several thermophiles have shown high potential for polymer degradation, similar to high-temperature plastic degradation agents. The bacteria can produce numerous enzymes with higher activity, which improves substrate bioavailability and solubility [67]. It was reported the first time that *Chelatococcus* sp. *E1* isolated from a compost sample was able to degrade PE. When pretreated PE samples were set at 60 °C, *Chelatococcus* sp. *E1* as thermophilic strains, PE co-cultured with *Chelatococcus* sp. *E1* apparently shifted the molecular weight distribution to the lower molecular weight side, increasing the biodegradability of HDPE and LDPE [42]. One of society’s main challenges may be solved by the development of extremozymes and the growth of extremophiles in severe environments.

## 5. Ways to Improve Biodegradation of Plastics/MPs

The pretreatment of plastic waste requires recycling or collection of microplastics in sewage in order to reduce the toxicity of additives, the adhesion of POPs, and depolymerized polymer chain [70]. The hydrophilicity of the polymer must be increased by chemical or biological oxidation processes in order to facilitate bacterial attachment and degradation. The results of various pretreatments can improve the efficiency of biodegradation. Recycling and biological processing achieve a significant reduction in plastic toxic pollution and improve biodegradation (Figure 4). However, considering further degradation of polymers, it is necessary to explore different characteristics of plastics for degradation. Biodegradable plastics are relatively more susceptible to pretreatment than traditional plastics due to their suitable physico-chemical properties, such as relatively low molecular weight, high flexibility, and more functional groups. Plastics can be pretreated with a variety of physical and chemical agents to alter their structural and morphological characteristics, such as reducing their molecular weight, rupturing chemical bonds, creating surface cracks, and enriching functional groups. All of these actions facilitate the biodegradation process that follows [32]. The changes in the biodegradation of plastics and MPs by different pretreatments should be determined, which can further increase the percentage of degradation. The pretreatment of plastic polymers with high temperature, photooxidation catalysis, and catalysis of microbial enzymes could promote their efficient biodegradation (Figure 4).

The pretreatment of PE, such as pyrolysis, makes it more easily metabolized by microorganisms [18]. Pretreatment using temperature for PET optimization improves the efficiency of PET-hydrolyzing enzymes, ultimately enhancing the degradation and temperature treatment. It is the most efficient pretreatment for PET degradation. The biodegradation activity of the enzymes, especially PETase, is affected by ambient temperature and PET structure. Due to the reduced flexibility of the PET molecules related to ambient temperature, the activity of the bacterial enzyme reached a higher value by suitable temperature, resulting in optimal catalysis. Other results show that the biodegradability by some bacteria of plastics that passed the temperature treatment is enhanced, and the broken or damaged effect of the corresponding enzymes on long-chain molecules of polymers is significantly increased **[25]**. For instance, the biodegradation of thermally pretreated HDPE by *Klebsiella pneumoniae CH001* was efficiently achieved, and the thermal pretreatment caused a significant reduction in the tensile strength of HDPE [32] (Figure 4).

By mixing with biodegradable additives, photo-initiators, or co-polymerization, weight reduction of PE degradation can be improved [71]. As a green way, photocatalysis has developed rapidly in recent years [72]. It was found that many simple iron salts can successfully catalyze the oxidative bond-breaking process of 400 nm LED irradiation lamp with pure oxygen or air, and ferrous chloride as the ideal photocatalyst [73]. Similarly, aromatic hydrocarbons, such as isopropylbenzene and cyclohexylbenzene, can undergo catalysis to achieve good C-C bond breaking and conversion to benzoic acid under the catalytic system [74]. The experiments confirmed that light, FeCl_3_, and air are necessary for efficient degradation (Figure 4). This method also could be efficiently used to break down commercial samples of high molecular weight PS [73].

The enzymatic surface modification of polymer fibers (i.e., changing the functional groups on the surface of plastic fibers) improves the wettability, fastness, dyeing, and pilling resistance of plastics, ultimately improving the hydrophilicity of plastics, such as lipase from *Candida antarctica* and keratinase from *Aspergillus oryzae*, in order to improve the hydrophilicity of fabrics [42]. With regard to plastics with C-C backbones, biodegradation using whole-cell enzymes is considered to be a potentially better method [75]. It is reported to use a multidisciplinary approach, monomer biocatalysis of PET, such as whole cell catalysis, becomes a biopolymer, which can form a biological cycle. Whole-cell catalysis can reduce the hydrophobicity of plastics and enhance the binding strength between enzymes and high crystalline substrate PET, with two basic steps including adsorption and hydrolysis [62]. As a whole-cell biocatalyzer, the anaerobic thermophile *Clostridium thermocellum* combines the synthesis of the enzyme with the hydrolysis of PET in one process (Figure 4). The result demonstrates that a PET film lost almost 60% of its weight over a 14-day incubation at 60 °C [76]. Thus, whole-cell catalysis can be one of the ideal additives for pretreatment catalysts [76]. The pretreatment of the various substrates using various parameters, such as photo-treatment and temperature, enzymatic pretreatment, and various additives, has indicated the enhancement of plastics and MPs biodegradation.

## 6. Conclusions

Plastic and MP pollution has been a long-standing problem with serious concerns in many fields because of its harmful effects on ecological and human health. There is currently a dearth of information on the capacity and potential of enzymes, including hydrolases, cutinase, and bacteriophilic enzymes, in the breakdown of plastics through enzymatic means. Therefore, further studies should be conducted to optimize effective enzymatic conditions for plastics degradation and focusing on the structural analysis of relative enzymes and the reaction mechanisms to obtain desired results. In recent years, many studies have focused on enzymes isolated from bacteria, such as PETase and cutinases. Therefore, selecting the suitable degradation microorganism for plastic treatment aiming at different types of plastics is of significance. Studies on the biodegradation of plastics can help to understand how microorganisms work and their potential to reduce the amount of plastic in the environment. They can also help to develop better enzymes to address the issues related to plastic waste. This review mainly concentrated on the breakdown of microorganisms in frequently used plastics and explored their enzymes, which operate as possible instruments for degradation that may greatly become the creation of sustainable bioremediation methods. Furthermore, studies on the impact of various pretreatment techniques and additives on the microbiological breakdown of synthetic plastics may produce better outcomes.

## Figures and Tables

**Figure 1 microorganisms-11-01661-f001:**
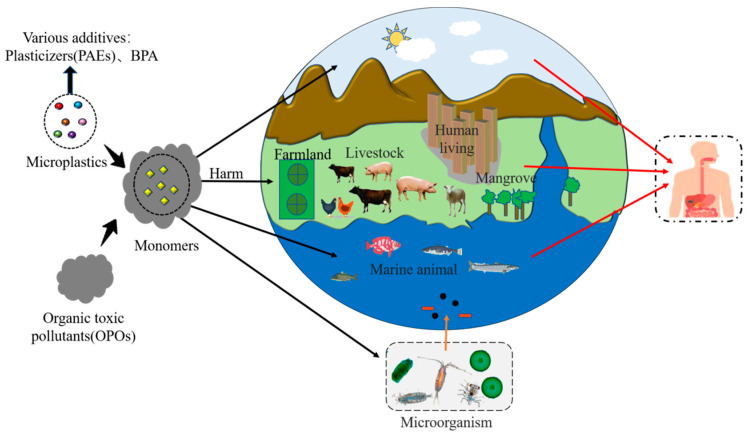
Additives, dyes, and organic pollutants in plastics can have negative effects on both humans and nature.

**Figure 2 microorganisms-11-01661-f002:**
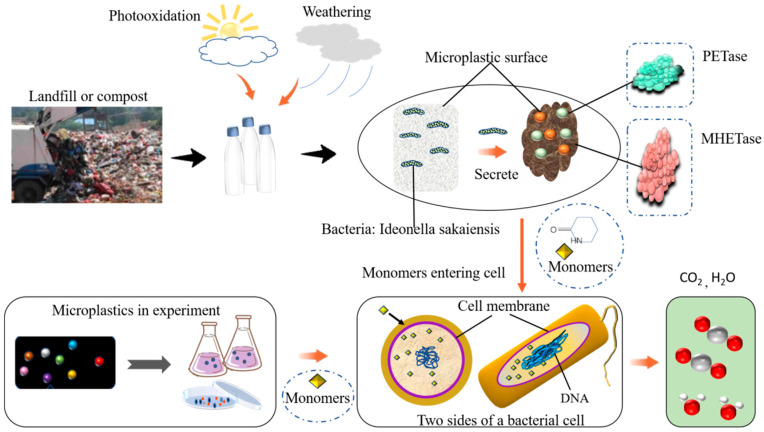
Degradation pathways of microplastics. Plastic degrading to small fragments forming microplastics can enter cells after decomposition. It can be transformed into biomass for energy production or mineralized.

**Figure 3 microorganisms-11-01661-f003:**
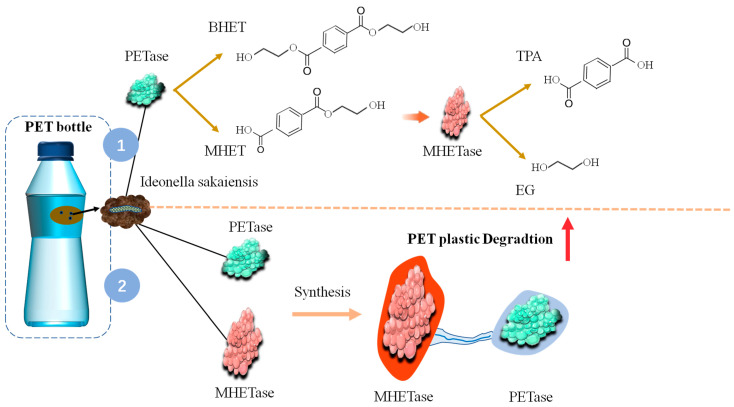
The enzymes involved in PET degradation secreted by *Ideonella sakaiensis*. *Ideonella sakaiensis* strain secretes enzymes PETase and MHETase, which degrade PET plastic, and the artificial enzyme PETase combined with MHETase significantly improve the degradation efficiency.

**Figure 4 microorganisms-11-01661-f004:**
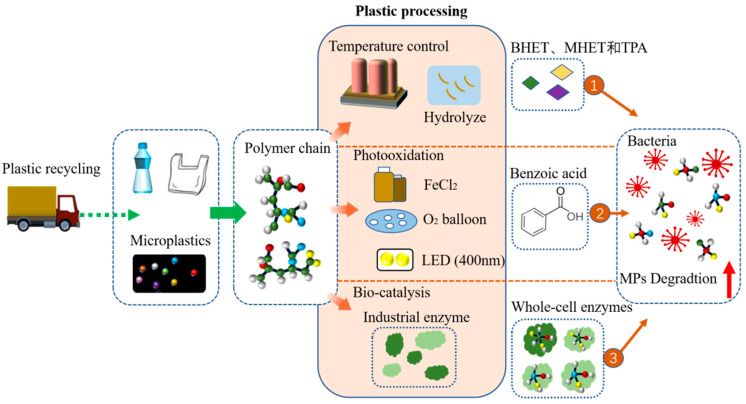
Pretreatment through physical and chemical methods can greatly make plastics prone to degradation.

**Table 1 microorganisms-11-01661-t001:** Enzymes and degradation factors corresponding to degrading bacteria.

Source	Enzyme	Major Mechanism of Degradation	Plastics/ Mircoplastics	Optimum Conditions	References
*Ideonella sakaiensis 201-F*	PETase	Hydrolysis	PET	Temperature 70–75 °C	[35]
*Ideonella sakaiensis 201-F*	MHETase	Hydrolysis	PET	Temperature 70–75 °C	[35]
*Amycolaptosis* sp.	——	Hydrolysis	PLA	Temperature 50 °C	[54]
*Aspergillus oryzae*	——	Hydrolysis	PBS	Temperature 50 °C	[15]
*Penicillium funiculosum*	——	Hydrolysis	PHB	Temperature 50 °C	[15]
*Thermomonospora and curvata* *(cutinase homolog from leaf-branch compost)*	LC-cutinase	Hydrolysis	PET	Temperature 50 °C	[65]
*Thermophilic, alkaliphilic, halophilic, and psychrophilic bacteria*	Bacteriophilic enzyme	Hydrolysis	Various plastic	Salt, low, or high pH, temperatures	[67]
*Pseudomonas, Arthrobacter*	PME hydrolases	Hydrolysis and oxidation	PVC, PP, PE, PS (PAEs)	Temperature 30–70 °C	[13]
*Bacillus* sp. *GZB*	A spore-laccase	The expression of different functional genes	PC (BPA)	Adding electron donors and co-substrates	[69]
*Aspergillus* sp. *strain ST-01*	Catalase, Protease	Colonization	PCL	Temperature 50 °C	[56]

## Data Availability

Not applicable.

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
