# Peer review of "Biological Degradation of Plastics and Microplastics: A Recent Perspective on Associated Mechanisms and Influencing Factors"

_microorganisms, 2023, doi:10.3390/microorganisms11071661_

Round 1

Reviewer 1 Report

The proposed review “Biological Degradation of Plastics and Microplastics: A Recent Perspective on Associated Mechanisms and Influencing Factors” submitted by Zeming Cai, Minqian Li, Ziying Zhu, Xiaocui Wang, Han Gong and Muting Yan summarizes studies on biological degradation of plastics with mechanisms and influencing factors, as well as their enzymes in enhancing the degradation of synthetic plastics in the process, with special focus on their modes of action and probable enzymatic mechanisms.

 The review is interesting and it can be published after minor revisions.

-        In the title and in the text are always mentioned plastics and microplastics, but for microplastics there are never specific references to studies carried out specifically on them.

-        Lines 33-34. The amount of plastics in 2018 includes fibers or not?

-        Some acronyms are not defined (ex. LDPE, HDPE, OTUs).

-        Line 71: typing error- plasticss

-        The captions to the figures are always texts of description and not schematic sentences. Please rewrite them.

-        Line 150: are viruses microorganisms?

-        Line 254: typing error- hydrolyz-able.

-        Line 275: (Fig. 3) Reference to the wrong figure

-        line 305: it is better to explain the principal differences between HDPE and LDPE.

-        Line 318: typing error: TIR spectra.

-        Line 330: typing error: Two capitol letter.

-        Line 384: cellulose, hemicellulose , and chitin are not plastics.

-        Lines 406 and 413. Reference to the wrong picture (Fig. 3).

-        Line 566: typing error: thermophile not italic.

-        Line 569: typing error: Whole capitol letter. 

Reviewer 2 Report

The discussion of the biological degradation of plastics is undoubtedly very important and relevant due to the ever-increasing pollution of the environment with plastics. However, there are many such reviews (Soong et al., 2022; Qi et al., 2022; Mohanan et al., 2020, etc.). Authors should clearly show how their review differs from that of other authors. In my opinion, they should add more specific data, now there are too many general phrases. In addition, the authors must improve their English, it is very difficult to read.

Some other comments:

1. In natural communities there are not only bacteria, but also fungi. What is known about the recycling of plastics by bacterial-fungal consortiums?

2. The authors do not mention Fusarium oxysporum. In this ascomycete, one of the first, cutinase, which depolymerizes PET, was described,.

3. Page 7: not only bacterial degradation of plastics is known, but also fungal.

4. Page 7, line 26: “microbial bacterium” is an unfortunate sentence.

5. PET is not degraded by Ideonella alone, but the authors discuss it only.

6. Page 9: The authors mention Ascomycetes, but not Fusarium. PET degradation has been described for F. oxysporum, F. solani, and some others.

7. Page 10: “Other types of fungi……. Penicillum spp." you need to add a reference.

 The authors must improve their English, it is very difficult to  read and understand.
